# Pluripotent Stem Cell-Derived Hepatocytes Inhibit T Cell Proliferation *In Vitro* through Tryptophan Starvation

**DOI:** 10.3390/cells11010024

**Published:** 2021-12-22

**Authors:** Marco Romano, Raul Elgueta, Daniel McCluskey, Ana Maria Ortega-Prieto, Emilie Stolarczyk, Francesco Dazzi, Baltasar Lucendo-Villarin, Jose Meseguer-Ripolles, James Williams, Giorgia Fanelli, David C. Hay, Fiona M. Watt, Giovanna Lombardi

**Affiliations:** 1Peter Gorer Department of Immunobiology, School of Immunology and Microbial Science, King’s College London, London SE1 9RT, UK; marco.romano@kcl.ac.uk (M.R.); raul.elgueta@mogrifuy.co.uk (R.E.); daniel.mccluskey@kcl.ac.uk (D.M.); giorgia.fanelli@kcl.ac.uk (G.F.); 2Department of Research and Development, Mogrify, Cambridge CB4 0FW, UK; 3Section of Virology, Faculty of Medicine, St Mary’s Campus, Imperial College London, London SW7 2AZ, UK; Ana.ortega.prieto@kcl.ac.uk; 4Division for Diabetes, Endocrinology and Metabolism, Faculty of Medicine, Hammersmith Campus, Imperial College London, London SW7 2AZ, UK; emilie.stolarczyk@kcl.ac.uk; 5Division of Cancer Studies, Rayne Institute, King’s College London, London SE1 9RT, UK; francesco.dazzi@kcl.ac.uk; 6Centre for Regenerative Medicine, Institute for Regeneration and Repair, University of Edinburgh, 5 Little France Drive, Edinburgh EH16 4UU, UK; balta.lucendo@gmail.com (B.L.-V.); jose.meseguer.ripolles@gmail.comuk (J.M.-R.); david.hay@ed.ac.uk (D.C.H.); 7Centre for Stem Cells & Regenerative Medicine, King’s College London, Guy’s Hospital, London SE1 9RT, UK; james.williams@kcl.ac.uk (J.W.); fiona.watt@kcl.ac.uk (F.M.W.)

**Keywords:** T cell activation, iPSC-derived hepatocyte-like cells, regenerative medicine, *in vitro* studies

## Abstract

Regenerative medicine aims to replace damaged tissues by stimulating endogenous tissue repair or by transplanting autologous or allogeneic cells. Due to their capacity to produce unlimited numbers of cells of a given cell type, pluripotent stem cells, whether of embryonic origin or induced via the reprogramming of somatic cells, are of considerable therapeutic interest in the regenerative medicine field. However, regardless of the cell type, host immune responses present a barrier to success. The aim of this study was to investigate *in vitro* the immunological properties of human pluripotent stem cell (PSC)-derived hepatocyte-like cells (HLCs). These cells expressed MHC class I molecules while they lacked MHC class II and co-stimulatory molecules, such as CD80 and CD86. Following stimulation with IFN-γ, HLCs upregulated CD40, PD-L1 and MHC class I molecules. When co-cultured with allogeneic T cells, HLCs did not induce T cell proliferation; furthermore, when T cells were stimulated via αCD3/CD28 beads, HLCs inhibited their proliferation via IDO1 and tryptophan deprivation. These results demonstrate that PSC-derived HLCs possess immunoregulatory functions, at least *in vitro*.

## 1. Introduction

Transplantation is the treatment of choice for many end-stage organs, such as the liver. However, organ rejection and long-term immunosuppression present a real challenge, and new strategies are needed [1]. One alternative strategy is the transplantation of allogeneic adult hepatocytes, which has been pioneered to reduce the severity of congenital errors of liver metabolism and chronic/acute failure to act as a bridge until the organ becomes available [2]. However, in both animals and humans, sustained liver function following hepatocytes’ transplantation has not been achieved [3]. Failure of cell engraftment, activation of the recipient’s immune response and the lack of a preconditioned environment play important parts in graft failure [4]. Hepatocytes from renewable and immune matched sources, such as pluripotent stem cells (PSCs), have been proposed as a concrete alternative to primary human hepatocytes to support organ function and regeneration.

Functional hepatocyte-like cells (HLCs) can be produced at scale from both embryonic stem cells (ESCs) and induced pluripotent stem cells (iPSCs) [5]. While autologous/syngeneic stem cell-derived products may lack immunogenicity once transplanted *in vivo*, stem cells generated from genetically different individuals are very likely to lead to the activation of the recipient’s immune response [6,7,8,9].

A pioneering study by Guha et al. tested the immunogenicity of terminally differentiated murine hepatocyte cells derived from ESCs and iPSCs *in vitro* and *in vivo* [10]. During co-cultures, a lack of T cell response was observed toward undifferentiated syngeneic iPSCs or their differentiated progenies. Consistent with the *in vitro* results, a lack of T cell infiltration was also observed following transplantation into syngeneic mice. More recently, Cisneros et al. showed that murine HLCs lacking MHC class I molecule expression were targeted *in vitro* by both syngeneic and allogeneic NK cells [11]. Lastly, in our previous work, human 3D liver tissue derived from ESCs provided critical liver support in two xenotransplantation models, including immunocompetent recipients, without being rejected [12].

To date, the immunogenicity of human HLCs remains poorly understood. Considering that autologous cell therapy would not be commercially viable, understanding the immunogenicity of allogeneic stem cell derived products is crucial for the development of future cellular therapies. Here, we investigated the phenotype of iPSC-HLCs and their potential to induce allogeneic T cell response *in vitro*.

## 2. Materials and Methods

### 2.1. HLCs Culture

Human iPSC-HLCs (iCellHeps 2.0) were obtained from Cellular Dynamics (Fujifilm Cellular Dynamics, Madison, WI, USA) from the Laboratory of Fiona Watts or Prof. D. Hay [13]. Human pluripotent stem cells were cultured using standard procedures [14]. In brief, hPSC maintenance was performed on pre-coated laminin-521 (Biolamina, Sundbyberg, Sweden) in mTeSR1 (STEMCELL Technologies, Vancouver, BC, Canada) at 37 °C, 5% CO_2_. HLCs differentiation was performed using standard procedures [15]. Briefly, hPSCs were seeded in a single-cell suspension at 50,000 cells/cm^2^. Differentiation was started when cells reached 40% confluence by replacing the medium with definitive endoderm differentiation medium RPMI 1640, containing B27 (Life Technologies, Carlsbad, CA, USA), 100 ng/mL Activin A (PeproTech, London, UK) and 50 ng/mL Wnt3a (R&D Systems, Minneapolis, MN, USA). The medium was changed every 24 h for 72 h. On day 3, endoderm differentiation medium was replaced by the hepatoblast differentiation medium, which was renewed every second day for 5 days. The medium consisted of KnockOut (KO) DMEM (Life Technologies, Carlsbad, CA, USA), Serum Replacement (Life Technologies, Carlsbad, CA, USA), 0.5% Glutamax (Life Technologies, Carlsbad, CA, USA), 1% non-essential amino acids (Life Technologies, Carlsbad, CA, USA), 0.2% β-mercaptoethanol (Life Technologies, Carlsbad, CA, USA) and 1% DMSO (Sigma-Aldrich, St. Louis, MO, USA ). At day 8, hepatoblasts were cultured for additional 10 days in the hepatocyte maturation medium consisting of HepatoZYME (Life Technologies, Carlsbad, CA, USA) containing 1% Glutamax (Life Technologies, Carlsbad, CA, USA) and supplemented with 10 ng/mL hepatocyte growth factor (PeproTech, London, UK) and 20 ng/mL oncostatin M (PeproTech, London, UK).

Adult human hepatocytes (Heps) were purchased from Life Technologies (Carlsbad, CA, USA) (Hu1651 and Hu8126).

Where indicated, HLCs as well as adult human hepatocytes have been treated with recombinant human IFN-γ (1000 U/mL, R&D Systems, Minneapolis, MN, USA) for 72 h.

### 2.2. T Cell Isolation and Proliferation

Anonymised healthy donor peripheral blood was obtained from the NHS Blood and Transplant Centre with informed consent and ethical approval (Institutional Review Board of Guy’s Hospital; reference 09/H0707/86). CD4^+^ or CD8^+^ T cells were enriched from ten different blood donors by using the RosetteSep™ Human CD4^+^ T Cell Enrichment Cocktail and the RosetteSep™ Human CD8^+^ T Cell Enrichment Cocktail, respectively (STEMCELL Technologies, Vancouver, BC, Canada).

For cell proliferation, CD4^+^ and CD8^+^ T cells were labelled with 5 µM CellTrace Violet Cell Proliferation Kit (Life Technologies Carlsbad, CA, USA) and cultured alone or at a 1:1 ratio with 0.5 × 10^5^ HLCs with or without αCD3/CD28-coated beads (ILife Technologies, Carlsbad, CA, USA) for 5 days (bead to cell ratio 1:40) in RPMI 1640 medium (Lonza, Basel, Switzerland) supplemented with 10% heat-inactivated FBS (Life Technologies, Carlsbad, CA, USA), 2 mM L-glutamine, 100 U/mL penicillin and 100 μg/mL streptomycin (MP Biomedicals, Santa Ana, CA, USA ) at 37 °C in 5% CO_2_. For testing the capacity of IDO1 in modulating T cell proliferation, where indicated, stimulated (αCD3/CD28-coated beads) and non-stimulated CD4^+^ T cells were co-cultured with IFN-γ-pre-conditioned HLCs in the presence/absence of 200 µM of 1-DMT, 1-LMT or tryptophan (all from Sigma-Aldrich, St. Louis, MO, USA).

Where blocking antibodies were used, HLCs were pre-treated for one hour with 5 µg/mL anti-human PD-L1 (clone: MIH1, eBioscience, San Diego, CA, USA) or Mouse IgG1 kappa Isotype Control (clone: P3.6.2.8.1 eBioscence, San Diego, CA, USA), anti-human CD40 (clone: 5C3, Biolegend, San Diego, CA, USA) or Ultra-LEAF™ Purified Rat IgG2b, κ Isotype Ctrl Antibody (clone: RTK4530, Biolegend, San Diego, CA, USA), anti-human IL-10R (clone: 3F9, Biolegend, San Diego, CA, USA) or Ultra-LEAF™ Purified Rat IgG2a, κ Isotype Ctrl Antibody (clone: RTK2758, Biolegend, San Diego, CA, USA) before the addition of T cells.

### 2.3. Flow Cytometry

The phenotypic analysis of HLCs was executed by using the following antibodies: CD86-PE (dil. 1:50, clone: IT2.2), CD40-APC-Cy7 (dil. 1:50, clone: 5C3), HLA-DR-Pacific Blue (dil. 1:50, clone: LN3), HLA-ABC-APC (dil. 1:50, clone: W6/32), CD80-BV605 (dil. 1:50, clone: 2D10) (all from Biolegend, San Diego, CA, USA) and PD-L1-PE-Cy7 (dil. 1:20, clone: MIH1 eBioscence, San Diego, CA, USA). The cognate isotype controls also used were PE Mouse IgG2b, κ (dil. 1:100 clone: 27–35 Biolegend, San Diego, CA, USA), Pe-Cy7 Mouse IgG1 κ Isotype Control (dil. 1:100, clone: P3.6.2.8.1 eBioscence, San Diego, CA, USA), APC-Cy7 Mouse IgG1, κ (dil. 1:100 clone: MOPC-21 Biolegend, San Diego, CA, USA), APC Mouse IgG2a, κ (dil. 1:100, clone: MOPC-173 Biolegend, San Diego, CA, USA), Pacific Blue Mouse IgG2b (dil 1:100 clone: MPC-11, Biolegend, San Diego, CA, USA) and BV605 Mouse IgG1, κ (dil 1:100, clone: MOPC-21, Biolegend, San Diego, CA, USA). Cells were incubated with Fc block for 20 min at 4 °C (dil. 1:100, Biolegend, San Diego, CA, USA) and then washed and stained with the antibody cocktail for other 20 min at 4 °C in PBS/EDTA (1 mM). Cell viability was measured by LIVE/DEAD™ Fixable Yellow kit (Life Technologies Carlsbad, CA, USA).

Intracellular staining for Albumin and IDO1 was executed after cell permeabilization with the Cytofix/Cytoperm™ Fixation/Permeabilization Solution Kit (BD biosciences, Franklin Lakes, NJ, USA) following manufacture instructions. Cells were labelled with IDO1-Pe (dil. 1:20 clone: eyedio) and Mouse IgG1 K Isotype Control PE (dil. 1:100) (both from ThermoFisher Scientific, Waltham, MA, United States) before acquisition. Goat anti-Human Albumin FITC (Bethyl Laboratories, Montgomery, TX, USA) was used in 1:200 with respective control (FITC-Goat IgG dil. 1:100).

For T cell proliferation, the cells were stained with anti-human CD3-PerCP (dil. 1:50, clone: OKT3), CD8-PeCy7 (dil. 1:50, clone: SK1), CD4-APC-Cy7 (dil. 1:50, clone: OKT4) and CD25-PE (dil. 1:50, clone: 4E3) from Biolegend (San Diego, CA, USA). For CD3ζ and FOXP3 evaluation, the FOXP3/Transcription Factor Staining Buffer Set (eBioscience, San Diego, CA, USA) was used following manufacturer’s instructions. The staining was performed with either Alexa Fluor 647 anti-human FOXP3 antibody (dil. 1:20, clone: 206D Biolegend, San Diego, CA, USA) or CD3ζ-PE (dil. 1:100 clone: 4B10) from eBioscience (San Diego, CA, USA). CellTrace Violet dilution was evaluated by flow cytometry at day 5.

Flow cytometry was performed on LSR Fortessa (BD Biosciences, BD biosciencies, Franklin Lakes, NJ, USA) and analysed with FlowJo software (v10.6.1 Tree Star).

### 2.4. qRT-PCR

Total RNA was isolated by using MiniKit RNeasy columns (QIAGEN) with a DNase-I treatment step. One microgram of DNA-free RNA was reverse transcribed to cDNA using Omniscript RT (QIAGEN, Hilden, Germany) following manufacturer’s instructions. TaqMan gene expression assays containing FAM dye-labelled TaqMan MGB probe were used for human HNF4a (Hs00230853_m1), SERPINA (Hs00165475_m1), Toll-like receptor (TLR) 2 (Hs01872448_s1), TLR4 (Hs00152939_m1), TLR5 (Hs01920773_s1), TLR7 (Hs01933259_s1) and TLR9 (Hs00370913_s1) in multiplex with VIC dye-labelled TaqMan probe for glyceraldehyde-3-phosphate dehydrogenase (GAPDH) as endogenous control. Real-time quantification (40 cycles) was performed using TaqMan Gene Expression Master Mix or Power SYBR Green PCR Master Mix (ThermoFisher, Waltham, MA, United States) on a Bio-Rad CFX96 optical reaction module on a C1000 thermal cycler following manufacturer’s instructions. Data were analysed using CFX Manager 3.1 Software (Bio-Rad, Hercules, CA, USA).

### 2.5. ELISA

Human serum albumin enzyme-linked immunosorbent assay (ELISA) was performed as previously described [16]. Human IL-10 (DY217B) and IFN-γ (DY285B) ELISAs (R&D Systems, Minneapolis, MN, USA) were used to evaluate the concentration of the two cytokines in the supernatant of the co-culture assays.

### 2.6. Statistical Evaluation

Results are expressed as mean ± SD. Comparisons between groups were performed using t-test for two groups or one-way ANOVA with a Dunnett or Bonferroni post-test for more than two groups. Analyses were performed using GraphPad Prism 8 software.

## 3. Results

### 3.1. Characterising iPSC-HLC Function and Immune Modulatory Molecule Expression

In order to understand the immunogenicity of iPSC-HLCs, we first evaluated whether specific markers of hepatocellular differentiation were maintained on HCLs in a pro-inflammatory microenvironment. We selected HNF4a, as it is a transcription factor essential for hepatocyte specification [17], and SERPINA, which is an alpha-1-antitrypsin enzyme produced specifically by hepatocytes [18]. At baseline, HNF4a and SERPINA gene expression in HLCs were similar to adult hepatocytes (Figure 1A,B), while undifferentiated iPSCs were negative for the expression of these two genes (Figure 1A,B). When HLCs were treated with IFN-γ, the gene expression of HNF4a and SERPINA was not modified. Similarly, albumin production remained unchanged following IFN-γ treatment in both HLCs and adult hepatocytes (Figure 1C,D). In addition, IFN-γ treatment did not affect the polyhedral shape of HLCs (Figure 1E). Taken together, these data suggest that HLCs’ differentiation state, function and morphology were not altered following IFN-γ treatment.

Due to the link with the gut via the portal vein, the liver is constantly exposed to gut-derived bacterial products. The detection of pathogen-associated molecular patterns (PAMPs) is mediated by the pattern recognition receptors (PRRs), including the Toll-like receptors (TLRs) that play a crucial role in maintaining hepatic immune homeostasis. Dysregulation in TLR signalling can lead to the production of proinflammatory cytokines and interferons favouring chronic liver diseases and fibrosis [19]. As human-cultured hepatocytes express the mRNA of all the TLRs at low levels [20], we aim to understand whether HLCs show the same expression pattern. HLCs express lower levels of TLR-2, -4, -5, -7 and -9 compared to adult hepatocytes, as shown in Appendix A.

We then evaluated the expression level of molecules involved in the activation/inhibition of cells contributing to adaptive immunity, namely CD4^+^ and CD8^+^ T cells. Specifically, we tested by flow cytometry the expression of HLA-ABC (major histocompatibility complex (MHC) class I), HLA-DR (MHC class II) and co-stimulatory/co-inhibitory receptors such as CD86, CD80, CD40 and PD-L1 before and after exposure to IFN-γ. HLCs constitutively expressed HLA-ABC with no expression of HLA-DR and co-stimulatory molecules (CD80 and CD86) (Figure 1F, Appendix A). Of note, IFN-γ treatment increased the levels of PD-L1, CD40 and HLA-ABC molecules (Figure 1F and G, Appendix A). 

Altogether, our results indicate that even after IFN-γ sensing, HLCs do not express the co-stimulatory receptors required to activate the T cells. 

### 3.2. Allogeneic iPSC-HLCs Inhibit the T Cell Immune Response

Recent reports have shown that different stem cell-derived products do not induce T cell proliferation *in vitro* [21,22]. Furthermore, another study has reported that adult human hepatocytes display immunosuppressive properties when co-cultured with human PBMCs [23]. To fully understand whether HLCs could induce an allogeneic immune response, CD8^+^ and CD4^+^ T cells were co-cultured with allogeneic HLCs treated or not with IFN-γ. HLCs did not induce CD8^+^ or CD4^+^ T cell proliferation (Figure 2A–C and Appendix A). Next, we evaluated whether they were able to suppress T cell proliferation following polyclonal activation. Indeed, HLCs were able to suppress the proliferation of both CD8^+^ and CD4^+^ T cells activated with αCD3/CD28 beads (Figure 2A–C and Appendix A).

The production of IL-10 and IFN-γ during the co-cultures was also analysed. We observed that IL-10 was increased in CD4^+^ T cells co-cultured with HLCs compared to CD4^+^ T cells cultured with αCD3/CD28 beads, although HLCs per se did not produce IL-10 (Figure 2D). In contrast, IFN-γ levels during the co-cultures were dramatically reduced compared to cultures of CD4^+^ T cells activated with αCD3/CD28 beads (Figure 2D). These results suggest that HLCs can induce a tolerogenic environment *in vitro* by increasing IL-10 production, suppressing T cell proliferation and their ability to release IFN-γ.

To evaluate whether the inhibition of T cell proliferation exerted by HLCs was dependent on cell-to-cell contact, HLCs treated or not with IFN-γ were separated from either CD8^+^ or CD4^+^ T cells by a transwell system. Untreated HLCs were able to suppress T cell proliferation during cell-to-cell contact, while this capacity was lost when the transwell system was used (Figure 2A–C). Of note, when T cells were co-cultured with HLCs treated with IFN-γ, T cell proliferation was inhibited even across the transwell insert. These findings imply that soluble inhibitory factors contributed to T cell suppression when HLCs were treated with IFN-γ.

To dissect the mechanisms behind the inhibition of cell proliferation in the presence of HLCs, neutralising antibodies specific to IL-10 receptor (IL-10R), CD40 and PD-L1 were used during the co-cultures. The results shown in Figure 2E demonstrated that both the expression of PD-L1 and CD40 by HLCs and the presence of IL-10 in the co-culture did not contribute to the inhibition of CD4^+^ T cell proliferation. This observation was extended to the co-culture of HLCs with CD8^+^ T cells to investigate whether the recognition of MHC class I molecules by CD8^+^ T cells could lead to different results from CD4^+^ T cells. However, the addition of blocking antibodies specific to PD-L1, CD40 and IL-10 did not revert the lack of T cell proliferation that HLCs mediated (Appendix A).

Another possible mechanism responsible for the inhibition of cell proliferation seen in our co-culture involves the conversion of conventional CD4^+^ T cells into regulatory T cells, as shown with iPSC-derived RPE cells [24]. A lower expression of FOXP3 was observed in stimulated T cells when cultured in the presence of iPSC-HLCs compared to T cells cultured with αCD3/CD28 beads only (Figure 2F and Appendix A). In addition, the presence of the different neutralising antibodies did not modify the levels of FOXP3 expression (Figure 2F and Appendix A).

### 3.3. HLCs Inhibit Allogeneic T Cell Proliferation by Tryptophan Starvation

Our results suggest that soluble factors may be involved in suppressing T cell proliferation mediated by IFN-γ pre-conditioned HLCs (Figure 2A–C). A previous work showed that the suppression of T cell proliferation by mesenchymal stem cells can be through tryptophan starvation mediated by indoleamine 2,3-dioxygenase-1 (IDO1) [25]. As IFN-γ is a potent IDO1 inducer, we hypothesised that HLCs may regulate T cell proliferation through this mechanism. Our results showed that the expression of IDO1 was increased following IFN-γ treatment (Figure 3A,B). To directly prove that IDO1 expression in HLCs was indeed involved in the suppression of T cell proliferation, the IDO1 inhibitor 1-L-methyl-tryptophan (1-LMT) and its non-functional isomer 1-D-methyl-tryptophan (1-DMT) were added to the co-cultures, and T cell proliferation was analysed. In the presence of the IDO1 inhibitor 1-LMT, but not 1-DMT, the suppressive effect of HLCs on CD4^+^ T cell proliferation was completely abrogated (Figure 3C,D). Moreover, supplementation of L-tryptophan (TRP) completely reversed the effect, with increased T cell proliferation observed (Figure 3C,E). Similar results have been found with CD8^+^ T cells (data not shown). These results suggest that IDO1 expression by HLCs contributes to T cell proliferation arrest.

Tryptophan deprivation induces a stress response, which results in downregulation of TCRζ chain in T cells, thus modulating their cell cycle and proliferation [26]. This was confirmed in our co-cultures as we observed a 2-fold reduction in the TCRζ chain expression (Figure 4A,B) in CD4^+^ T cells co-cultured with HLCs treated with IFN-γ. Additionally, we observed a reduction in the expression of CD25 molecules, a T cell activation marker (Figure 4C). In contrast, in the presence of either TRP or 1-LMT to counterbalance the IDO1 effect, the TCRζ chain and CD25 expression were re-established (Figure 4B,C). Thus, IDO1-mediated tryptophan starvation is a mechanism used by HLCs to downregulate the TCRζ chain and CD25 molecules in CD4^+^ T cells, which lead to T cell proliferation arrest.

## 4. Discussion

Liver transplantation represents the treatment of choice for several end-stage liver failures. However, the organ availability, as well as the detrimental side effects caused by immunosuppression, reduce the outcome of this life-saving procedure. So far, different strategies have been adopted to increase organ acceptance and induce operational tolerance [27,28,29]. Compared to other organs, the liver has an extraordinary capacity to regenerate; therefore, cell-based regenerative therapies are expected to become an attractive alternative strategy to organ transplantation in the near future. Functional hepatocyte-like cells (HLCs) derived from allogeneic sources can be produced in large numbers from both embryonic stem cells (ESC) and induced PSC (iPSC) and have the potential to be an off-the-shelf product. However, similarly to organ transplantation, allogeneic HLCs can activate the host immune system, causing rejection and graft loss. In this study, we have shown that HLCs do express MHC class I molecules but not MHC class II or CD80/CD86 molecules. Furthermore, they upregulate MHC class I molecules, CD40 and PDL1 in the presence of IFN-γ. Finally, these cells were not immunogenic *in vitro* and showed immunoregulatory capacity by inhibiting T cell proliferation via tryptophan starvation.

In solid organ transplantation, the recognition of donor antigens by recipient T cells via direct/indirect and semi-direct allorecognition triggers a strong immune response [30]. This occurs through the recognition of MHC class I and class II molecules, expressed on either donor or recipient antigen-presenting cells by recipient CD8^+^ and CD4^+^ T cells, respectively. Similarly to other stem cell-derived products, HLCs constitutively express MHC-I molecules that are upregulated following IFN-γ exposure [21,31]. Although these cells express MHC class I molecules, CD8^+^ T cells were not activated, likely due to the lack of co-stimulatory molecules expression (Figure 1F,G and Figure 2A). However, MHC class I molecules expressed by HLCs can be presented either indirectly or semi-directly following cell infusion, and specific *in vivo* models are required to test this scenario. To reduce the immunogenicity of both ESC- and iPSC-derived cells, different groups are now generating cell products lacking MHC class I molecules. However, the lack of these molecules can trigger recipient NK cell activation, causing graft loss [31]. To bypass this problem, RPE and hESC neural progenitor cells have been engineered to overexpress MHC molecules acting as NK inhibitors, such as HLA-E, HLA-F and HLA-G [32,33,34,35].

The cell product evaluated in this study did not express HLA-DR even when the cells were exposed to IFN-γ. This represents an advantage over other cellular therapies, such as RPE cells, which, by upregulating HLA-DR following IFN-γ treatment, acquire the ability to directly present the antigens to CD4^+^ T cells [21,31].

HLCs expressed the co-inhibitory molecule PD-L1, which is further upregulated following IFN-γ. A constitutive expression of PD-L1 by MSC contributed to suppressing CD4^+^ T cell activation and IL-2 secretion [36]. Furthermore, the upregulation of PD-L1 following IFN-γ stimulation has been also found in ESC-RPE, and it has been proposed as a possible mechanism mediating T cell suppression [21]. The suppressive role of PD-L1 has been also shown *in vivo* by Yoshihara et al. This group overexpressed PD-L1 on human islet-like cells, ameliorating their survival and function compared to non-engineered cells [37]. However, in our experimental model, PD-L1 did not play a role, as demonstrated by the lack of an effect on the inhibitory function of HLCs when neutralizing anti-PD-L1 antibodies were added to the system (Figure 2F and Appendix A).

Here, we have shown that HLCs have an immunomodulatory function, as they inhibited T cell proliferation. This was not directly due to IL-10 production or Treg induction (Figure 2 and Appendix A). In our system, IDO1-mediated tryptophan starvation downregulated the levels of TCRζ chain and CD25 molecules in CD4^+^ T cells, which lead to T cell proliferation arrest. IDO-mediated immunomodulatory effect has already been described in mesenchymal stem cells derived from different compartments [25,38] and human iPSC-derived mesoangioblasts [39]. Furthermore, IFN-γ priming enhanced the immunosuppressive properties of human MSCs due to the upregulation of IDO1 [40]. Whether the immunomodulatory capacity of the HLC can be reverted by danger signals, as shown by Zhang et al. with liver sinusoidal endothelial cells [41], is still a matter of debate. Based on our data, HLC-mediated immune tolerance may not be affected by the ligation of TLRs; however, new functional data are necessary to evaluate whether and how HLC respond to TLR stimulation.

In conclusion, our *in vitro* findings provide some mechanistic understanding of T cell immunomodulation via IDO1 expression and tryptophan starvation by stem cell-derived HLCs. This mechanism may be important for inducing local immune suppression when the cells are infused *in vivo*. In the future, when reliable humanised models of liver injury and HLC engraftment are established, the immunoregulatory capacities of these cells will be fully examined *in vivo*.

## Figures and Tables

**Figure 1 cells-11-00024-f001:**
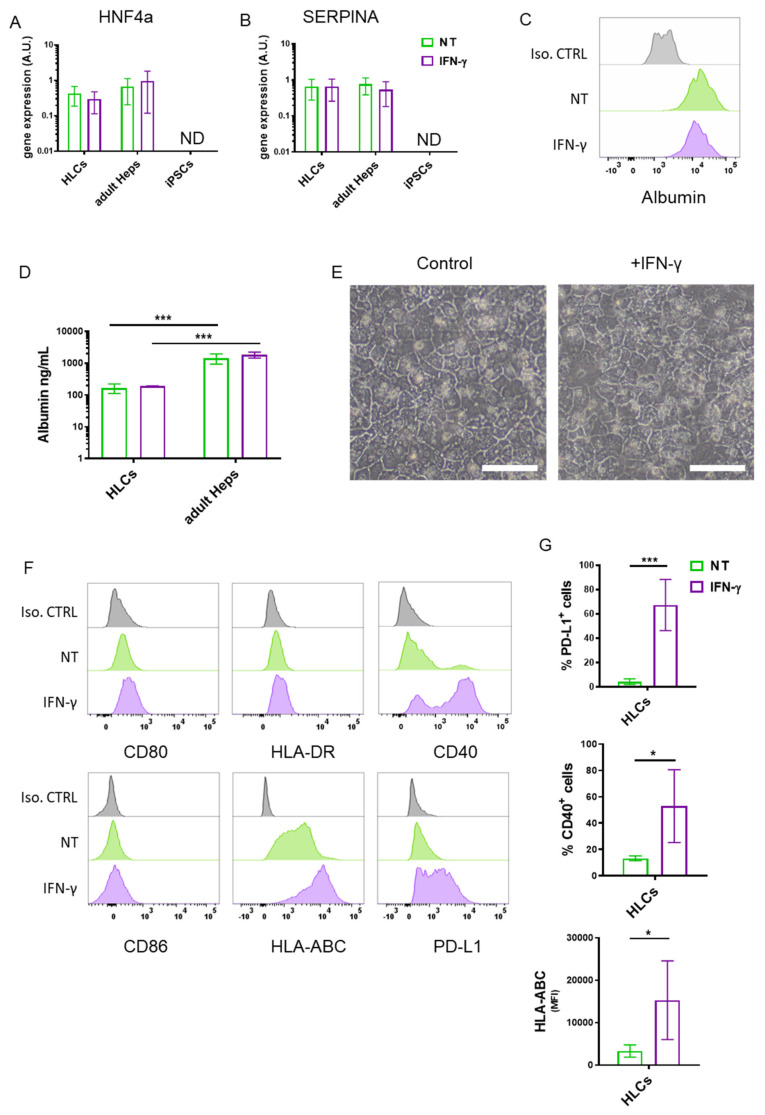
Immune phenotyping of HLCs. (**A**) HNF4a and (**B**) SERPINA gene expression in HLCs, adult hepatocytes (Heps) and non-differentiated iPSCs treated or not with IFN-γ (1000 U/mL) for 72 h. (**C**) Representative flow cytometry histogram of albumin expression in HLCs treated or not with IFN-γ (1000 U/mL). (**D**) Albumin production analysed by ELISA in HLCs and adult hepatocytes (Heps) treated or not with IFN-γ (1000 U/mL) for 72 h. (**E**) Representative image of HLCs treated or not with IFN-γ. Scale bar represents 100 µm. (**F**) Representative histograms of CD80, CD86, HLA-DR, HLA-ABC, CD40 and PD-L1 expression analysed by flow cytometry. Groups: non-treated HLC (NT) (green), IFN-γ-treated HLCs (IFN-γ) (purple) and isotype (grey). (**G**) Quantification of the percentages of positive cells for CD40 and PD-L1 and evaluation of the MFI of HLA-ABC (lower graph) in HLCs treated or not with IFN-γ. * *p* < 0.05, *** *p* < 0.001. (**D**) Two-way ANOVA. (**G**) Student’s *t*-test. N.D.: not detected. Data are expressed as mean ± SD of 6 independent experiments, with one donor per experiment, in duplicate.

**Figure 2 cells-11-00024-f002:**
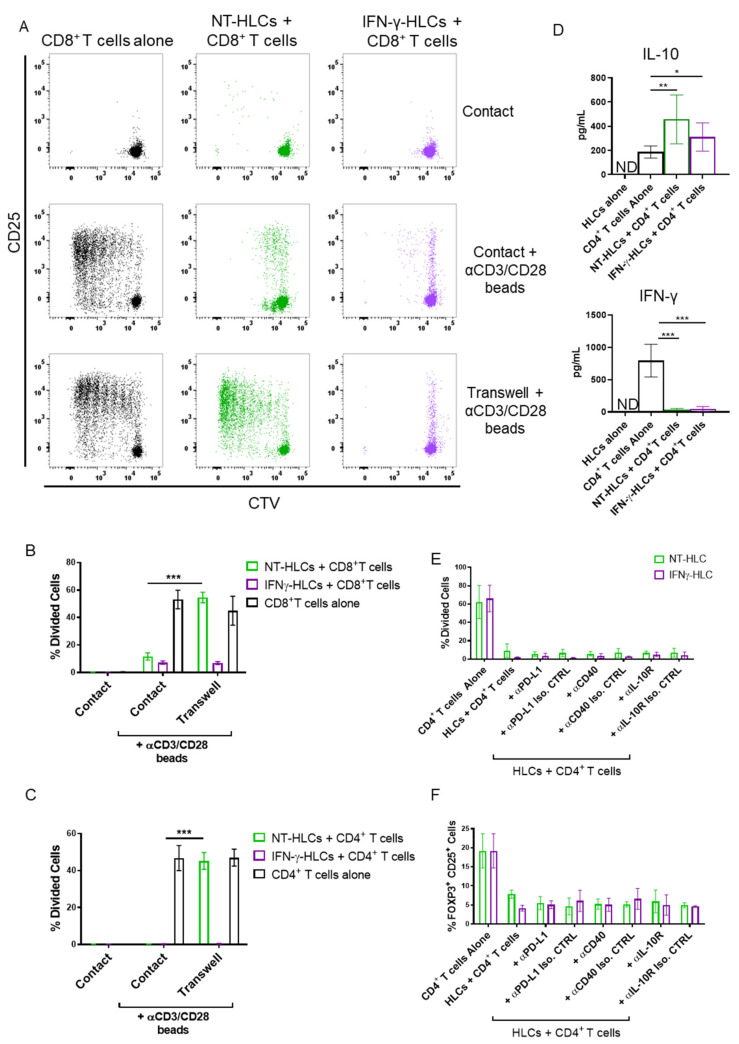
Allogeneic HLCs inhibit T cell activation. CellTrace Violet (CTV)-labelled CD8^+^ or CD4^+^ T cells were co-cultured for 5 days with or without αCD3/CD28 beads (bead to cell ratio 1:40) in the absence (alone) or in the presence of HLCs treated or not with IFN-γ (1000 U/mL) in the same wells (contact) or separated by transwells. (**A**) Representative dot plots of CD8^+^ T cell proliferation and upregulation of CD25 expression. (**B**) Quantification of the percentage of divided CD8^+^ T cells co-cultured with HLCs. (**C**) Quantification of the percentage of divided CD4^+^ T cells co-cultured with HLCs. (**D**) IL-10 (upper panel) and IFN-γ (lower panel) production from co-cultures of HLCs and CD4^+^ T cells. (**E**) Quantification of the percentage of divided CD4^+^ T cells activated with αCD3/CD28 beads and co-cultured with HLCs in the presence of αPD-L1, αCD40, αIL-10R and cognate isotype controls. (**F**) percentage of CD25^+^FOXP3^+^ in CD4^+^ T cells activated with αCD3/CD28 beads and co-cultured with HLCs in the presence of αPD-L1, αCD40, αIL-10R-blocking antibodies and cognate isotype controls. * *p* < 0.05, ** *p* < 0.01, *** *p* < 0.001. (**B**,**C**) Two-way ANOVA. (**D**,**E**) One-way ANOVA. N.D.: not detected. Data are expressed as mean ± SD of 4 independent experiments, with one donor per experiment, in duplicate.

**Figure 3 cells-11-00024-f003:**
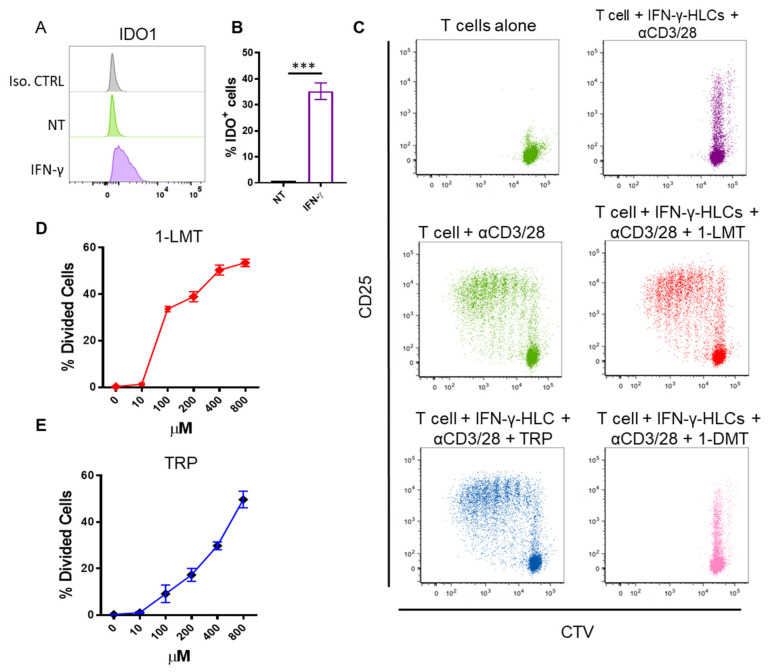
IDO1 expression in allogeneic HLCs suppress T cell proliferation by tryptophan starvation. (**A**) Representative histogram showing IDO1 expression in HLCs. In grey is isotype control, in green non-treated (NT) HLCs and in purple IFN-γ-treated HLCs. (**B**) Cumulative data showing the expression of IDO-1 in HLCs treated (median = 35.25 range = 4.5) or not with IFN-γ (median = 0.735, range = 0.013). (**C**) Representative dot plots of CellTrace Violet-labelled CD4^+^ T cells, co-cultured for 5 days with or without αCD3/CD28 beads (bead to cell ratio 1:40) in the absence (alone) or in the presence of IFN-γ-treated HLCs in the presence of 1-LMT, tryptophan or 1-DMT (200 µM). Percentage of divided CD4^+^ T cells co-cultured with HLCs treated with IFN-γ in the presence of different doses of 1-LMT (**D**) and tryptophan (TRP) (**E**). Data are expressed as mean ± SD of 3 independent experiments with one donor per experiment *** *p* < 0.001.

**Figure 4 cells-11-00024-f004:**
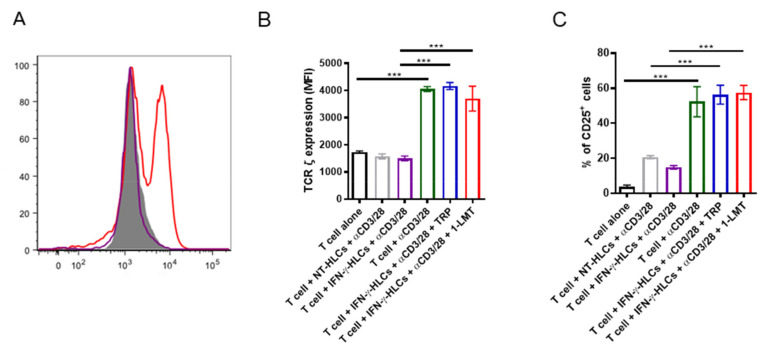
TCRζ expression in CD4^+^ T cells. (**A**) Representative histogram for TCRζ expression in CD4^+^ T cells co-cultured with IFN-γ-treated HLCs and in the presence (red line) or absence (purple line) of 200 µM 1-LMT. Isotype control in grey. (**B**) Quantification of TCRζ expression (MFI) in non-activated (median = 1728 range = 216) and activated (median = 4023 range = 175) CD4^+^ T cells co-cultured with IFN-γ-treated HLCs (median = 1527 range = 167) in the presence of 200 µM of 1-LMT (median = 3923 range = 826) or tryptophan (median = 4171 range = 311). (**C**) Percentage of CD25^+^ cells in non-activated (median = 3.7 range = 1.8) and activated (median = 52 range = 17) CD4^+^ T cells co-cultured with IFN-γ-treated HLCs (median = 14.7 range = 2.5) in the presence of 200 µM of 1-LMT (median = 57.2 range = 8.1) or tryptophan (median = 56.65 range = 10.4). *** *p* < 0.001. One-way ANOVA. Data are expressed as mean ± SD of 3 independent experiments, with one donor per experiment, in duplicate.

## Data Availability

The data that support the findings of this study are available from the corresponding author, [G.L.], upon reasonable request.

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
