# Peer review of "Pluripotent Stem Cell-Derived Hepatocytes Inhibit T Cell Proliferation In Vitro through Tryptophan Starvation"

_cells, 2021, doi:10.3390/cells11010024_

Round 1

Reviewer 1 Report

General Comments

The study of Romano, Elgueta et al. investigated the in vitro immunogenicity of induced pluripotent stem cell-derived hepatocytes (iPSC-HLCs) by evaluating their effect on T-cell proliferation before and after interferon-g (IFN-g) treatment. Data showed that iPSC-HLCs act via tryptophan starvation and IDO-1 expression describing information that could be of interest not only in the field of organ transplantation, but also in the context of acute and chronic inflammatory diseases such as acute coronary syndromes or cancer.

Comments

  • Data are expressed as mean ± SD and statistical analyses were performed using tests for parametric data. Are all data distributed normally? Please, show both sides of the error bars in each histograms.
  • The authors used isotype as flow-cytometry controls. Did the authors acquire unstained cells for evaluating auto-fluorescence, especially on treated cells?
  • T cells were isolated from human blood with RosetteSep. Authors should complete the purification method by detailing the post-enrichment step.
  • Please describe the cell culture steps (conditions, passages, etc.) detailing the reagent used too.
  • Figure legends: please, check the symbol g of IFN-g, and m of mm throughout the manuscript.
  • The quality of the figures should be improved. Please, check the pixel resolution.
  • Each graph should have titles in x- and y-axes (for example, the flow cytometry plots).
  • Check the Ref at line 113.
  • Line 289: reference number is missing.

Reviewer 2 Report

Summary

Stem cells could be a good source as an alternative to live tissue transplantation, especially for liver tissue. In this manuscript, authors investigated whether liver cells derrived from induced pluripotent stem cells have immunostimulatory and/or modulatory functions, as immune activation in vivo is not wanted in the case of allogeneic transplantations/transplantations derrived from a collection of stem cell lines expressing defined HLA molecules. Authors found that liver cells derived from stem cells rather induce immune tolerance in T cells, and further investigate how this is mediated for CD4 T cells. I think the work is interesting, the evidence provided is suitable, but some details are lacking in the manucsript and these need to be adressed before this manuscript can be considered for publication.  Furthermore, I believe the involvement of danger signals such as TLRs in T cell priming/activation, especially in the context of the liver, have been shown profusely in the past and warrant investigation in this context. Lastly, while the content of the discussion seems OK, some references are missing, sentences are off, or some interesting typos were made, making it seem like the authors did not pay a lot of attention to detail in this section. At some point, I decided not to list all textual changes anymore as they began to be too numerous.

Major comments

  • The materials and methods section is lacking a lot of information. No information regarding IFN-g treatment or co-cultures is given. Similarly, no information regarding the CD3/CD28 controls (types of antibodies, # of ug, duration of stimulation) is provided. Please provide information on all assays employed in this manuscript in the materials and methods. This information is critical and needs to be added in detail.

Suggested additions

  • TLR ligands have the potential to lower TCR thresholds in murine and human T cells (e.g. doi: jimmunol.1801026). Other danger signals have also shown to function in this capacity (i.e. STING, cytokines). Previous research has shown that liver-cell induced CD8 T cell tolerance is counteracted in the presence of TLR ligands (e.g. doi: s41423-019-0255-8). It would be interesting if authors could corroborate this, or whether this does not play a role in HLCs, especially as TLRs and in particular danger signals from dying cells can be abundantly present in vivo during infections and immune responses, and could thus influence also HLC-mediated immune tolerance. Do HLCs express relevant TLRs or PRRs? Are they functional? Can triggering counteract the tolerance induction?

Minor/textual comments

Line 52 – “issues” and “products” (2x add the s)

Line 53 – ”stem cells” (add the s)

Line 55 – please provide a reference to any of the older transplantation studies where this was studied for completeness.

Line 58 – “responses” (add the s)

Line 67 – “allogeneic stem cell-derived product” (remove the s in cells)

Line 74 – What does 1818 mean after the name of Prof. Hay? Should this refer to a paper where the cells were described? Please correct.

Line 87 – please add ref number + manufacturer for isotype control antibodies.

Line 88 – please add dilution of FoxP3 antibody

Line 92 and onward– please add dilution of all used antibodies

Line 97 – please add Fc block manufacturer and concentration/dilution used, and in what medium/buffer this was performed.

Line 97 – please provide buffer flow cytometry staining was performed in.

Line 99 – please correct name of LIVE/DEADTM kit and add the fluorophore (Near-IR or Orange for instance).

Line 100 - please provide FlowJo version.

Line 104 – please add details for reverse transcription and taqman gene expression, or specify that experiment was performed according to manufacturer’s instructions.

Line 107 – please provide information about cycling.

Line 110 – please provide CFX Manager Software version.

Line 113 – please correct reference.

Line 117 – Please provide GraphPad Prism version.

Line 123 – please correct double spaces.

Line 127 – no information regarding IFN-g treatment is present in materials and methods. Please add.

Line 139 –  I don’t agree with this statement – there is an increase in the percentage of cells that expresses CD40 (as evident by the + peak), and an increase in MHC-I MFI (as evident by the peak shift to the right). Please provide MFI information in the figure for both (maybe even for all markers tested here) and re-state this sentence. Ideally, authors should provide bar graphs for all markers tested, for both % & geoMFI. The statement about PD-1 is semi-correct – IFN-g treatment resulted in higher amounts of PD-L1+ hepatocytes than HLCs.

Line 143 – not entirely correct either, also here I would ask authors to restate as, at this point in the manuscript, this point has not been made yet. Something more appropriate would be “Altogether, our results indicate that, even after IFN-g sensing, HLCs do not express the co-stimulatory receptors required to activate T cells.”

Line 145 – please correct gamma character in figure legend, do this for all subsequent figure legends please.

Figure 3 – please correct the indicating letters here, A-E are behind the panels. Furthermore, please provide information regarding the amount of donors included in this experiment and/or repeats that have been performed. If this is only performed once, this has to be repeated.

Line 237 – please correct Zeta sign, do so throughout (also in discussion for instance).

Figure 4 – how long was this culture, or control activation?

Discussion – some sentences don’t really seem finished or correct, such as line 288 – the sentence about PD-L1 is cut short, please correct. Have a good look at the discussion and fix typographical, grammatical and other issues throughout. Interestingly, the discussion is the first part where authors use CD4+ (add the plus sign), which is the preferred writing style for many authors/journals. Please make the use of + consistent throughout the manuscript (preferably adding it to both CD4 and CD8 throughout). Line 289 is missing the reference, just says REF.

Please note, the content of the discussion is fine/good, but needs to be double checked and finished, as the discussion is unpublishable in it's current state.

Round 2

Reviewer 2 Report

Dear authors,

Thank you for incorporating most (if not all) of my suggestions and adding the TLR expression data. However, now it seems like a random addition; therefore, it would be good if the context (re: TLR-mediated priming of cells residing in the liver and the role thereof in T cell activation, e.g. doi 10.1038/s41423-019-0255-8) would be briefly mentioned in the discussion, as that would explain why this was added and complete the story. Indeed, as stated in your cover letter, some extra experiments would be insightful in this case to understand whether danger signals could revert this immunomodulatory capacity of HPC-derived hepatocytes, and that can be mentioned in the discussion. Besides that, I have 5 very minor points that could be addressed, see below.

----

Line 74 – there is a space missing before reference [14]

Line 116 + 117 – 2x remove CD8+

Section starting at line 136 – please be consistent in the order in which clone and dilution is mentioned

Line 144 – remove _ after FlowJo software

Line 184 – TLRs are not the only sensors of PAMPs, also NLRs and other danger receptors fullfill this role; I would suggest to re-state to say “The detection of PAMPs by is mediated by pattern recognition receptors such as TLRs” to be inclusive.
